# A Two-Stage Neural-Filter Pareto Front Extractor and the need for Benchmarking

## Abstract

Pareto solutions are optimal trade-offs between multiple competing objectives over the feasible set satisfying imposed constraints. Fixed-point iterative strategies do not always converge and might only return one solution point per run. Consequently, multiple runs of a scalarization problem are required to retrieve a Pareto front, where all instances converge. Recently proposed Multi-Task Learning (MTL) solvers claim to achieve Pareto solutions combining Linear Scalarization and domain decomposition. We demonstrate key shortcomings of MTL solvers, that limit their usability for real-world applications. Issues include unjustified convexity assumptions on practical problems, incomplete and often wrong inferences on datasets that violate Pareto definition, and lack of proper benchmarking and verification. We propose a two stage Pareto framework: Hybrid Neural Pareto Front (HNPF) that is accurate and handles non-convex functions and constraints. The Stage-1 neural network efficiently extracts the *weak* Pareto front, using Fritz-John Conditions (FJC) as the discriminator, with no assumptions of convexity on the objectives or constraints. An FJC guided diffusive manifold is used to bound the error between the true and the Stage-1 extracted *weak* Pareto front. The Stage-2, low-cost Pareto filter then extracts the *strong* Pareto subset from this *weak* front. Numerical experiments demonstrates the accuracy and efficiency of our approach.

## 1 Introduction

Multi-Objective Optimization (MOO) problems arise frequently across diverse fields such as engineering (Marler & Arora, 2004), finance (Tapia & Coello, 2007), and supply chain management (Trisna et al., 2016). Such problems share a common requirement to satisfy multiple competing objectives under a set of constraints imposed by physical or economic limits. A Pareto optimal solution Pareto (1906) for an MOO problem is defined as the solution point away from which no single objective can be improved without diminishing at least one other objective. A Pareto front is then defined as the set of all such optimal points that satisfy this definition.

Since practical MOO problems contain competing objectives and additional domain specific constraints, a Pareto solver should be able to handle both functions and constraints. Moreover, problems in classification and recommendation are non-convex (Hsieh et al., 2015), necessitating solvers that are robust for non-convex problems. Although many saddle point solvers have been proposed in the literature (Benzi et al., 2005; Benzi & Wathen, 2008), they cannot be generalized to handle constraints without specialized pre-conditioners designed to handle non-linear constraints. These existing solvers are based upon fixed-point iterations, returning one solution per run with specialized local initialization to generate an even spread of Pareto points across the feasible set of saddle points.

Recently proposed Multi-Task Learning (MTL) solvers (Sener & Koltun, 2018; Lin et al., 2019a; Mahapatra & Rajan, 2020; Ma et al., 2020; Navon et al., 2021) criticize Linear Scalarization (LS) since it retrieves a subset of the Pareto optimal set. MTL solvers claim to resolve this issue by decomposing the functional/variable domain (into cones or rays) while using LS in each of the sub-domains. However, the stationary points of the LS problem (now decomposed) remain unchanged (see **Appendix B**), and therefore any fixed-point iterative strategy will still retrieve only a subset of the Pareto optimal set. Additionally, the theorems presented in MTL works rely upon convexity assumption that are never justified through numerical experiments. Furthermore, a notable absence of benchmarks against known analytical forms makes it difficult to assess reported results, verify

optimality, and A/B test alternative methods. This is in contrast to studies in Operations Research (OR) (Das & Dennis, 1998; Ghane-Kanafi & Khorram, 2015; Pirouz & Khorram, 2016) in which such comparative benchmarking and verification is well established. Although accurate and verifiable, existing OR methods tend to generate Pareto points with low density and limited scalability, with compute times ranging from hours to days as variable dimensionality increases.

While lack of formal guarantees for non-convex cases certainly does not preclude use of MTL methods for such problems, rigorous numerical evaluation becomes particularly important. However, the true Pareto front (closed-form analytical solution) is often unknown for real MOO problems, hence it is challenging to compare the accuracy of a Pareto solver on such problems. We follow the OR literature in advocating that the correctness of any proposed Pareto solver should first be tested on constructed benchmark problems with known analytic solutions. This is also consistent with broader ML community practice of first evaluating proposed methods in controlled synthetic or simulated conditions to verify correctness, before applying these systems on real datasets. In general, rigorously evaluating a new method on a diverse set of complicated benchmarking scenarios can provide valuable insights prior to venturing into the unknown and builds user trust in the model.

Table 1: HNPF *vs.* existing OR and MTL methods. OR methods are accurate and can handle constraints, with limited scalability for high-dimensional Neural MOO problems. MTL approaches scale well but do not support constraints or provide benchmark comparisons against known analytic solutions for non-convex problems.

| Method | Pareto points only | Even Spread | Handle Constraints | Neural Scalable | Manifold Solution |
|---|---|---|---|---|---|
| mCHIM (Ghane-Kanafi & Khorram, 2015) | ✓ | ✓ | ✓ | ✗ | ✗ |
| PK (Pirouz & Khorram, 2016) | ✓ | ✓ | ✓ | ✗ | ✗ |
| NBI (Das & Dennis, 1998) | ✓ | ✓ | ✓ | ✗ | ✗ |
| **HNPF** | ✓ | ✓ | ✓ | ✗ | ✓ |
| MOOMTL (Sener & Koltun, 2018) | ✗ | ✗ | ✗ | ✓ | ✗ |
| PMTL (Lin et al., 2019a) | ✗ | ✗ | ✗ | ✓ | ✗ |
| EPO (Mahapatra & Rajan, 2020) | ✗ | ✗ | ✗ | ✓ | ✗ |
| PHN (Navon et al., 2021) | ✗ | ✗ | ✗ | ✓ | ✓ |

We propose a two-stage framework: Hybrid Neural Pareto Front (HNPF) for extracting Pareto optimal solution sets. **Stage 1** consists of an interpretable neural network that extracts a *weak* Pareto solution manifold as the output, given a dataset as input. Following this, **Stage 2** is a low-cost Pareto filter to remove dominated points from the *weak* Pareto set. The network loss function uses a discriminator based on **Fritz-John Conditions (FJC)** to account for multiple objectives and constraints. An approximate *weak* Pareto manifold is extracted as a weighted output of the *softmax* function from the last layer of the network. The softmax activation classifies *weak* Pareto *vs.* non-Pareto data points. HNPF extracts this *weak* Pareto front as a continuous manifold approximated by the Stage-1 neural network. Numerical experiments show the computational efficiency of HNPF *vs.* OR approaches and accuracy comparisons against MTL methods. HNPF produces only Pareto points (no false positives) with an even spread and high density. Furthermore, HNPF is scalable (compared to OR methods) with both increasing dimensions of the variable domain, and the number of functions and constraints. **Table 1** summarizes key properties of HNPF *vs.* existing methods. Our key contributions are:

1. A neural manifold extraction strategy for *weak* Pareto front identification based on Fritz-John conditions as the discriminator, for both convex and non-convex scenarios, supporting constraints.

2. The final neural net layer is interpretable as a continuous approximation of the *weak* Pareto manifold. The extracted manifold error is bounded below *w.r.t.* the true manifold upon convergence.

3. The necessity of Pareto filter to remove dominated points from the *weak* Pareto set. Design of a computationally efficient Pareto filter to extract the non-dominated Pareto optimal set.

4. Raising community awareness about existence of benchmarks from literature in OR field.

## 2 PROBLEM STATEMENT

A general MOO problem can formulated as:
$$optimize \quad F(x) = (f_1(x), f_2(x), \ldots, f_k(x)) \tag{1}$$

$$s.t. \quad x \in \mathcal{H} = \{x \in \mathbb{R}^n | G(x) = (g_1(x), g_2(x), \ldots, g_m(x) \leq 0\}$$

in $n$ variables $(x_1, \ldots, x_n)$, $k$ objective functions $(f_1, \ldots, f_k)$, and $m$ constraint functions $(g_1, \ldots, g_m)$. Here, $\mathcal{H}$ is the feasible set *i.e.* the set of input values $x$ that satisfy the constraints $G(x)$. For a MOO problem there is typically no single global solution, and it is often necessary to determine a set of points that all fit a predetermined definition for an optimum. See **Appendix A** for definitions.

## 2.1 FRITZ JOHN CONDITIONS (FJC)

Let the objective and constraint function in Eq. 1 be differentiable once at a decision vector $\tilde{x}^* \in \mathcal{H}$. The Fritz-John necessary conditions (FJC) for $\tilde{x}^*$ to be *weak* Pareto optimal is that vectors must exists for $0 \leq \lambda \in \mathbb{R}^k$, $0 \leq \mu \in \mathbb{R}^m$ and $(\lambda, \mu) \neq (0, 0)$ (not identically zero) *s.t.* the following holds:

$$\sum_{i=1}^{k} \lambda_i \nabla f_i(\tilde{x}^*) + \sum_{j=1}^{m} \mu_j \nabla g_j(\tilde{x}^*) = 0, \quad \mu_j g_j(\tilde{x}^*) = 0, \forall j = 1, \ldots, m \qquad (2)$$

Gobbi et al. (2015) presented an $L$ matrix form of FJC as follows:

$$L = \begin{bmatrix} \nabla F & \nabla G \\ \mathbf{0} & G \end{bmatrix} \quad [(n+m) \times (k+m)] \qquad (3)$$

$$\nabla F_{n \times k} = [\nabla f_1, \ldots, \nabla f_k], \quad \nabla G_{n \times m} = [\nabla g_1, \ldots, \nabla g_m], \quad G_{m \times m} = diag(g_1, \ldots, g_m)$$

In $L$ matrix, for $x^*$ to be Pareto optimal, is to show the existence of $\delta \in \mathbb{R}^{k+m}$ in Eq. 2 such that

$$L \cdot \delta = 0 \quad \text{s.t.} \quad L = L(\tilde{x}^*), \delta \geq 0, \delta \neq 0 \qquad (4)$$

The non-trivial solution ($\delta$ not identically zero) for Eq. 4 is:

$$det(L^T L) = 0 \qquad (5)$$

**Remark.** *Note that $det(L) = 0$ is equivalent to $det(L^T L) = 0$ (see **Appendix G** for derivation).*

The *weak* Pareto front is characterized by the set of points such that matrix $L$ is low rank. This ensures that points identified are either inside the feasible set or at boundaries dictated by the constraints. For *e.g.* if $\mu_1 = 0$ for any $\lambda_i$, then $\sum_i \lambda_i f_i = 0$ must be satisfied for the corresponding internal point $x^*$ to be Pareto. Similarly if $\mu_1 \neq 0, \mu_{j \neq 1} = 0$ in the aforementioned case, then $g_1 = 0$ holds true for the corresponding boundary point $x^*$ to be Pareto. All Pareto points satisfy $\nabla f_i = 0$ for at least one $i$ whether they lie inside the feasible set or on the boundary, *i.e.* all points $x^*$ need to be local optimizers (stationary points)[1] for at least one $f_i$. The rank of the matrix $L^T L$ determines the dimension of the Pareto manifold. FJC written as $det(L^T L)$ is independent of the preference parameters $\lambda_i$, $\mu_j$. Thus, **Eq. 5** is an oracle serving as a discriminator in HNPF to identify a *weak* Pareto front.

**Remark.** *For a multi-objective optimization (MOO) problem scalarization results in a single-objective optimization (SOO) problem. However, the stationary points of this resulting SOO problem can only be a small subset of the weak Pareto solution set. The reader is referred to **Appendix B** for a visual explanation of stationarity w.r.t. an SOO problem arising from an MOO problem.*

## 3 RELATED WORKS

**Generic and Enhanced Scalarization:** One common approach is to convert an MOO problem into a Single Objective Optimization (SOO) problem via Linear Scalarization (LS). These include: Balashankar et al. (2019); Lin et al. (2019a); Martinez et al. (2020); Valdivia et al. (2021); Wei & Niethammer (2020). Enhanced scalarization approaches (Das & Dennis, 1998; Ghane-Kanafi & Khorram, 2015; Pirouz & Khorram, 2016) rely upon localization of the objective space to handle non-convex functions and constraints. Although accurate and complete, these approaches suffer from low computational scalability and low density of Pareto points on the solution manifold. For example, the 30 dimensional benchmark (**Case V** in **Section 5**) shows enhanced scalarization methods (mCHIM and PK) generating a Pareto set in approximately 18 hours.

**Multi Task Learning:** MTL methods rely upon a utopia/ideal point (similar to Das & Dennis (1998)) but are shown to be scalable for high-dimensional MOO problems. MOOMTL (Sener & Koltun, 2018) uses a multi-gradient descent approach, but does not guarantee an even spread of solution points along the Pareto front. PMTL (Lin et al., 2019a) attempts to address this issue by dividing the functional domain into equal spaced cones, with increasing computational cost as the number of cones increase. EPO (Mahapatra & Rajan, 2020) uses preference rays along specified weights to find Pareto points in the vicinity of the rays. EPSE (Ma et al., 2020) uses a combination Hessian of the functions and Krylov subspace to find Pareto solutions. PHN (Navon et al., 2021) is initialized either by LS or EPO and is the first neural framework that can approximate the front as a neural manifold.

---

[1]A stationary point *w.r.t.* gradient descent fixed point iteration is the set of points corresponding to the local or global minima, maxima or saddle points of an objective function.

The reader is further referred to **Appendix O** for a detailed review of these methods including other from the Bayesian, Genetic and Fairness literature.

**Current Limitations:** Firstly, these aforementioned point-based approaches extract one solution point at a time, given the optimization problem converges. However, for practical applications, with non-convex objectives and constraints, ensuring optimality is non-trivial. Secondly, multiple runs with different trade-off parameters must be performed in order to extract the *weak* Pareto solution set, resulting in substantial computational overhead. Finally, the Pareto solution set can still form a non-convex manifold even when the objectives are convex due to the presence of non-convex constraints (**Case III** in **Section 5**). These challenges prove to be major obstacles in the deployment of MOO solution methods as practical tools for Pareto set extraction.

## 4 HNPF Framework

We draw inspiration from three seminal works: **1)** Das & Dennis (1998) proposed to break the functional domain boundary into uniform and evenly spaced segments to identify *weak* Pareto points with guarantees. Motivated by this, we first identify the *weak* Pareto front using a neural network (Stage 1). **2)** Messac et al. (2003) proposed the first Pareto filter to obtain the set of strong Pareto points from the aforementioned *weak* Pareto set. The filter uses an all-pair comparison criterion to reject dominated points from the *weak* Pareto set. This motivates our low-cost Pareto filter (Stage 2) using a plane search strategy to avoid an expensive all-pair comparison. **3)** Gobbi et al. (2015) presented a matrix form of the Fritz-John conditions satisfying the existence of *weak* Pareto points.

### 4.1 Stage 1: Neural Net for Weak Pareto Front

Given FJC, one can brute force classify the data points as *weak* Pareto or not. However, our intent is to extract an approximate Pareto front as an indicator function ($\tilde{M} : \mathbb{R}^n \rightarrow \mathbb{R}$) parameterized by the neural network. The Stage-1 neural network (**Fig. 1**) smoothly approximates the true *weak* Pareto manifold $M(X^*)$ as $\tilde{M}(\tilde{X})$ (see **Appendix F** for details on notation). The last layer has two neurons with *softmax* activation for binary clas-

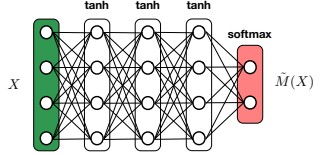

Loss = Binary Cross Entropy $\left( det \left( L(X)^T L(X) \right), \tilde{M}(X) \right)$

Figure 1: HNPF Stage-1 with FJC guided binary cross entropy loss to extract *weak* Pareto front.

sification of Pareto *vs.* non-Pareto points, distinguishes *weak* Pareto *vs.* sub-optimal points, in the feasible set $\mathcal{H}$. Our network loss is representation driven, since the FJC guided discriminator (**Eq. 5**), described by the objectives and constraints, explicitly classifies the input data points $X$ points.

The Pareto front is a *lower dimensional manifold* ($\leq (n-1)$) in an $n$-dimensional variable domain. The Pareto *vs.* non-Pareto classes are not comparable since the measure of the Pareto front is zero. In other words, the volume of a surface (or anything lower) is zero. To volumize the Pareto front and to make a Binary Cross Entropy measure computable, we use a *diffusive indicator function*. The volume can then be made tending to zero by choosing a smaller $\epsilon$ approximating the lower dimensional weak-Pareto manifold. We use a slightly relaxed criterion with user-tunable threshold $0 \leq \epsilon \leq 1$ as the classification margin. Any point below this value will be classified as *weak* Pareto. The FJC for *weak* Pareto optimality then requires that the $D = det(L^T L) = 0$. Therefore, $|D| \leq \epsilon$ (True or False) naturally provides us with binary labels for the *softmax* activated output layer. A binary cross entropy loss ensures that the distribution of the extracted manifold $\tilde{M}(\tilde{X})$ matches the distribution of the *weak* Pareto front satisfying FJC within the specified $\epsilon$ tolerance. This is similar to recently popularized Physics Informed Neural Networks (PINNs) (Raissi et al., 2019) where *a known analytical form of the regression function is supplied as a loss term, instead of explicit labels*.

**Error bound and Network Interpretability.** For a user-prescribed relaxation margin $0 \leq \epsilon \leq 1$, the approximation error between the network extracted manifold $\tilde{M}(\tilde{X})$ and the true solution $M(X^*)$ is bounded below by $\|\tilde{M}(\tilde{X}) - M(X^*)\|_2 \leq \epsilon$, upon convergence. See **Appendix F** for proof and **Appendix H** for a discussion on interpretability as a mathematical model.

**Remark.** *For a $n$-dimensional variable domain, HNPF (Stage 1) uses the Fritz-Johns conditions in the determinant form to extract a $n$-dimensional, diffusive, indicator function using a binary cross entropy loss (weak Pareto vs. non-Pareto). The indicator function regresses to 1 at the weak Pareto front and 0 everywhere else with values between 0 and 1 in the $\epsilon$ neighborhood of this weak Pareto front (diffusive indicator). In this respect, Fritz-Johns condition is an oracle to gather the weak Pareto set where the determinant form is used instead of user provided labels to identify the front.*

## 4.2 STAGE 2: PARETO FILTER FOR STRONG PARETO SET

A Pareto filter is an algorithm that, given a set of *weak* Pareto points $\mathcal{P}$ of cardinality $|\mathcal{P}| = P$ in objective space, retains a subset of non-dominated points. This corresponds to the strong Pareto set *s.t.* none of the points are dominated, *i.e.* the filter eliminates all dominated points from $\mathcal{P}$. The state of the art Pareto filter (Messac et al., 2003) requires an all-pairs comparison ($\mathbf{O(P!)}$ time complexity), that becomes computationally expensive as the set $\mathcal{P}$ grows in size. However, since Stage-1 of HNPF generates *weak* Pareto points with high density, the filter proves to be quite expensive.

**Remark.** *A fixed point iterative approach might converge to the stationary points (local criterion) of an MOO problem. However, it is not necessary that all such stationary points are non-dominated. A filter is therefore necessary to remove the dominated points based upon a global criterion. See Section 5.1 and 5.4 for numerical experiments where a filter is necessary.*

We present an efficient Pareto filter for finding the strong Pareto set which is computationally scalable to arbitrary dimensions. The algorithm is based on a plane search strategy, inspired by Kd-Trees (Bentley, 1990), well known for efficient data partitioning and storage. The compute cost is guided by the number of functions while being linearly proportional to the number of points. The inputs to the algorithm (**Fig. 2**) are the number of functions $k$ and their respective global minima and maxima. These are known *a priori* for benchmark problems, or can be computed from the data itself in $O(kP)$ time *i.e.* a linear search over the $k$ functions for $P$ points. $f(min)$, $f(max)$,

```
1:  Data 𝒫 = {x̃*} ∈ ℝᵏ weak Pareto points
2:  Input fᵢ(min), fᵢ(max);    ∀i ∈ k
3:  Input k : #functions, h : discretization level
4:  for i ∈ k do
5:      level = fᵢ(min)
6:      for j ∈ (fᵢ(max) − fᵢ(min))/h do
7:          temp = ∅
8:          for p ∈ 𝒫 do
9:              if level ≤ fᵢ(p) < level + h then
10:                 temp = temp ∪ p
11:             if card(temp) > 1 then
12:                 xₚ = min f_q(x), x ∈ temp, q = i + 1
13:                 𝒫 = 𝒫\(temp\xₚ)
14:             level = level + h
15: Output: Non-dominated Pareto set 𝒫 = {x*}
```

Figure 2: Pareto Filter Algorithm

discretization level $h$ of the function space and the *weak* Pareto points $P$. The output is the strong Pareto set $\{x^*\}$. Refer to **Appendix N** for algorithmic details and usage illustration of the filter.

**Time complexity:** The three nested *for* loops carry the load. In the worst case that the points in the *weak* Pareto set are all strong Pareto, then the cardinality $P$ of the set $\mathcal{P}$ remains unchanged. Let $z = (f_i(max) - f_i(min))/h$ denote the number of chunks into which the function space is divided. The worst case complexity of the proposed Pareto filter is $\mathbf{O(kzP)}$. For scenarios, where the strong Pareto set is a subset of the original *weak* set $\mathcal{P}$, the complexity reduces in the factor guided by $P$.

## 4.3 COMPARISON BETWEEN HNPF AND PHN

HNPF (Stage-1) extracts the *weak* Pareto manifold as an $n$-dimensional diffusive indicator function as opposed to a $(n-1)$-dimensional manifold itself. This results in an indicator function $\tilde{M} : \mathbb{R}^n \to \mathbb{R}$ that with an $n$-dimensional support in the variable space. *There are two explicit advantages in describing this indicator function:* (**1**) The regressed Pareto manifold is not only guided by the weak Pareto points (indicator value 1) but also the sub-optimal points (indicator value 0) for a more robust and accurate extraction. This is comparable to classification under a balanced class density wherein if the class densities were increasingly imbalanced, the extracted classification boundary becomes more inaccurate. (**2**) The Pareto optimal set can be extracted and represented by a neural network (function approximator) even when the manifold is an implicit surface. For eg., consider representing a circle (equation as opposed to a function) as network extracted manifold in a 2D variable domain.

In comparison, PHN (Navon et al., 2021) uses a neural network to regress over the $(n-1)$-dimensional manifold in the variable domain directly, given solution points obtained from EPO or LS. However, i) neither EPO nor LS are guaranteed to work for non-convex scenarios (see Section 5); ii) if the non-dominated Pareto set is discontinuous, then PHN will wrongly identify non-Pareto points as Pareto optimal; iii) even when the Pareto optimal set is a continuous manifold, PHN inherently assumes that the point spread obtained from EPO or LS is uniformly distributed over the true manifold for the neural regressor to not overfit; and iv) if the manifold is an implicit surface then a direct regression using a neural network is not feasible. Further numerical justifications can be found in **Appendix D**.

## 5 RESULTS

In this section, we present four numerical experiments for benchmarking and analysis, with five more in **Appendix I**. These address analytical forms, with increasing complexity and scale in the number

of *functions* $(k)$, *constraints* $(m)$ and variable *dimension* $(n)$. The benchmark cases are chosen to test various aspects desired from a general Pareto solver. We compare HNPF *vs.* two OR methods: mCHIM and PK[2]; and four recent MTL methods: MOOMTL, PMTL, EPO, and PHN. For HNPF's setup and error tolerance specification, refer to **Appendix J**. Additional experiments are presented as an ablation study (**Appendix K**) with loss profiles (**Appendix L**) and runtimes (**Appendix M**).

**Data Sampling:** Since the location of the Pareto front is not know *a priori*, samples are drawn uniformly at random from a feasible variable domain. Objective functions evaluated at these points generate a quantized, topographic map of the function domain that is then used to identify optimal points. For each benchmark test case below, we generate **11k** points from a random uniform distribution in the feasible variable domain, to serve as training data. The training-validation split is **90-10%**. Once the manifold $\tilde{M}(\tilde{X})$ is approximated by the network, we use **90k** points in the feasible domain as test data to visualize the Pareto set.

**Remark.** *Although MTL methods do not explicitly handle constraints, the Pareto solution set is only reported for the benchmark cases (I, II, IV) where the linear constraints form a bounding box.*

### 5.1 Challenges in Pareto Optimality & the Need for a Pareto Filter $(n = 2, k = 1, m = 0)$

We now consider a two objective, unconstrained, MOO problem in a single variable domain where the two, non-convex objective functions are given by:

$$f_1(x) = (x - 1)(x - 2)(x - 3)(x - 4)$$
$$f_2(x) = (x - 1.5)(x - 2.5)(x - 3.6)(x - 4.5)$$

We design this benchmark MOO problem to specifically test out HNPF's robustness when multiple optimal manifolds exist for the corresponding SOO problem. The LS single-objective function is:

$$S(\alpha, x) = \alpha f_1(x) + (1 - \alpha) f_2(x)$$

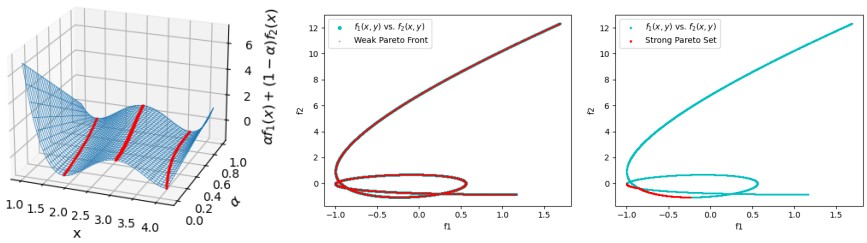

(a) LS Single Objective    (b) HNPF *weak* Pareto set    (c) HNPF *strong* Pareto set

Figure 3: Pareto front extraction for the triple optimal manifold benchmark problem.

**Fig. 3 (a)** shows the surface of $S(\alpha, x)$ in $(\alpha, x)$ (blue mesh) with the three optimal manifolds (red curves). Note that for any point $x$ there exists at least one $\alpha$ on one of the three manifolds. Therefore, the final Pareto set can be formed by a union of the subset of each of these three manifolds. **Fig. 3 (b)** shows that the optimal manifolds (red curve) overlaps the entire $f_1$ *vs.* $f_2$ plot (cyan curve) in the functional domain, as expected. The Pareto optimal set is then obtained by applying a filter based upon the global criterion of non-dominated points in the functional domain as shown in **Fig. 3 (c)**.

Note that for this particular benchmark case, the Pareto optimal set is a union of the subset of two optimal manifolds in **Fig. 3 (a)**. All MTL solvers fail in this case when the Pareto front in the functional domain is an implicit, self intersecting manifold. This numerical experiment also enunciates the importance of a Pareto filter to satisfy the global criterion of non-dominated points inherent to the definition of Pareto optimality. *A fixed point iteration converges locally by satisfying a local optimality/termination criterion. The global criterion of non-dominated points can only be satisfied once the optimal set is gathered completely from the previous step.*

---

[2]Since the codes for mCHIM and PK are not publicly available, we snip results from their works to avoid any artifacts in reproducibility. Hence the color discrepancy in plots.

## 5.2 CASE I: CONVEX OBJECTIVES, LINEAR CONSTRAINTS ($n = 2, k = 2, m = 2$)

This problem was originally proposed in Fonseca & Fleming (1998). Jointly minimize:

$$f_1(x_1, x_2) = 1 - exp(-[(x_1 - 1/\sqrt{(2)})^2 + (x_2 - 1/\sqrt{(2)})^2])$$
$$f_2(x_1, x_2) = 1 - exp(-[(x_1 + 1/\sqrt{(2)})^2 + (x_2 + 1/\sqrt{(2)})^2])$$
$$\text{s.t.} \quad g_1, g_2 : -1/\sqrt{2} \leq x_1, x_2 \leq 1/\sqrt{2}$$

*This is a common benchmarking problem in both OR and machine learning literature.* Since the objectives and constraints are convex, HNPF (see **Appendix I.1**), OR and MTL methods all retrieve the Pareto optimal set accurately. The reader is referred to **Appendix B** for a detailed discussion and visual explanation of Pareto optimality using a variant of this problem.

## 5.3 CASE II: NON-CONVEX OBJECTIVES, LINEAR CONSTRAINTS ($n = 2, k = 2, m = 2$)

This problem was proposed in Ghane-Kanafi & Khorram (2015). Jointly minimize:

$$f_1(x_1, x_2) = x_1, \quad f_2(x_1, x_2) = 1 + x_2^2 - x_1 - 0.1 sin 3\pi x_1$$
$$\text{s.t.} \quad g_1, g_2 : 0 \leq x_1 \leq 1, -2 \leq x_2 \leq 2$$

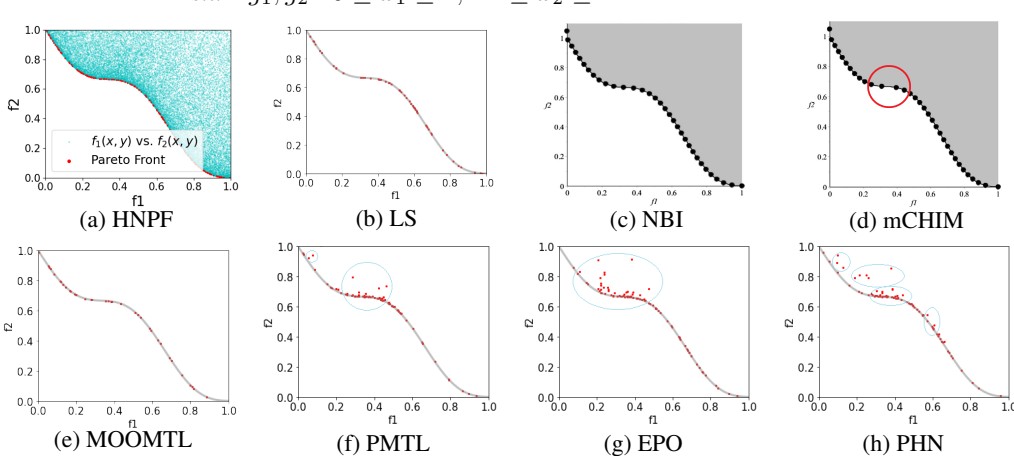

(a) HNPF    (b) LS    (c) NBI    (d) mCHIM

(e) MOOMTL    (f) PMTL    (g) EPO    (h) PHN

Figure 4: Pareto Front for Case II. Note the even spread of point HNPF produces. NBI produce an even spread while mCHIM cannot (uneven spread in red circle). PMTL and EPO produces sub-optimal points (blue circle). Since PHN relies on EPO as a solver, it correspondligly fails in regions where EPO fails.

HNPF, LS, NBI, mCHIM, and MOOMTL methods identify Pareto points accurately while PMTL, EPO, and PHN also generate sub-optimal points (marked blue circle). **Fig. 4** shows the results from our model with high point density. It also satisfies closely the true Pareto manifold $M(X^*)$ given by $0 \leq x_1 \leq 1, x_2 = 0$ in Fig. 4(b).

## 5.4 CASE III: CONVEX OBJECTIVES, NON-CONVEX CONSTRAINTS ($n = 2, k = 2, m = 4$)

This problem was proposed in Tanaka et al. (1995). Jointly minimize:

$$f_1(x_1, x_2) = x_1, \quad f_2(x_1, x_2) = x_2$$
$$\text{s.t.} \quad g_1(x_1, x_2) : (x_1 - 0.5)^2 + (x_2 - 0.5)^2 \leq 0.5$$
$$g_2(x_1, x_2) : x_1^2 + x_2^2 - 1 - 0.1 \cos(16 \arctan(\frac{x_1}{x_2})) \geq 0$$
$$g_3, g_4 : 0 \leq x_1, x_2 \leq \pi$$

Here, $f_1, f_2$ are convex but the constraints $g_1, g_2$ are non-convex in the variable domain and consequently the resulting Pareto front is non-convex. An important point to note here is that the Pareto front is dominated solely by the two constraints $g_1$ and $g_2$, while linear functions $f_1$ and $f_2$ do not contribute. *Since MTL solvers do not account for constraints, a comparison is only shown against OR methods that report results for this benchmark.* Here, mCHIM extracts a sparse set of non-dominated Pareto points ($\sim 40$) while HNPF Stage-1 extracts a dense set of *weak* Pareto points (**Fig. 5**). To arrive at the non-dominated Pareto set, we post-process this result using our Stage-2 Pareto filter (see Section 4.2). The updated discontinuous set of non-dominated Pareto points is shown in Fig. 5 (b).

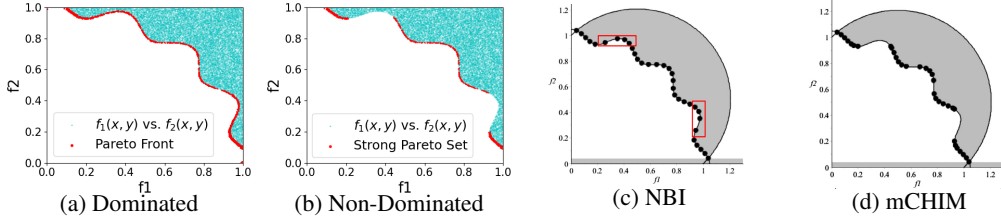

Figure 5: Pareto Front for Case III. (a) HNPF weak front in function space. (b) All dominated points are removed from the set after applying the Pareto filter.

## 5.5 CASE IV: NON-CONVEX OBJECTIVES, LINEAR CONSTRAINTS ($n = 30, k = 2, m = 30$)

This problem was proposed in Zhang et al. (2008). Jointly minimize:

$$f_1(x) = x_1 + \frac{2}{|J_1|} \sum_{j \in J_1} y_j^2 \quad , \quad f_2(x) = 1 - \sqrt{x_1} + \frac{2}{|J_2|} \sum_{j \in J_2} y_j^2$$

$$\text{s.t.} \quad g_1, \ldots, g_{30} : 0 \leq x_1 \leq 1, -1 \leq x_j \leq 1, j = 2, \ldots, m$$

$$J_1 = \{j | j \text{ is odd}, 2 \leq j \leq m\}, J_2 = \{j | j \text{ is even}, 2 \leq j \leq m\}$$

$$y_j = \begin{cases} x_j - [0.3x_1^2 \cos(24\pi x_1 + \frac{4j\pi}{m}) + 0.6x_1]\cos(6\pi x_1 + \frac{j\pi}{m}) & j \in J_1 \\ x_j - [0.3x_1^2 \cos(24\pi x_1 + \frac{4j\pi}{m}) + 0.6x_1]\cos(6\pi x_1 + \frac{j\pi}{m}) & j \in J_2 \end{cases}$$

This form is non-convex in both $f_1, f_2$. The dimension of the design variable space is $m = 30$. The Pareto front in the variable domain is non-convex (sinusoidal spiral), implicit surface. HNPF results (Fig. 6) are in good agreement with mCHIM and PK methods, but with significantly higher density. See **Appendix M** for compute time scalability of HNPF *vs.* OR methods. *All MTL methods fail with the inherent min-norm solver returning NaN.*

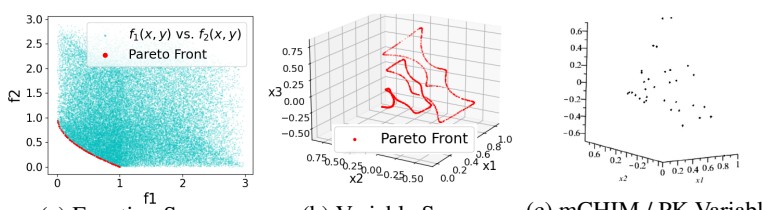

Figure 6: Pareto Front for Case IV. (a, b) HNPF front for Function and Variable space respectively. Note the density difference between HNPF and mCHIM / PK in the variable space.

## 5.6 DISCUSSION OF METHODS

**OR methods:** NBI works for cases where the detected *weak* Pareto front consists of non-dominated points only. Therefore, NBI generates correct solution in Cases I, II, IV and V with even density of points on the Pareto front. In essence, applying the Pareto filter on the NBI generated solution set would resolve the discontinuous cases too. NBI, mCHIM, PK and HNPF produce only Pareto points. Additionally, HNPF generates Pareto points uniformly with high density, while OR methods including mCHIM and PK, although accurate, are limited to low point density ($\sim 40$) with large compute overhead as the variable dimension scales. **Table 2** shows a comparison of the extracted Pareto point density (#extracted Pareto points/#function evaluations) for HNPF, OR, and MTL methods.

Table 2: Extracted Pareto point density (100*#extracted Pareto points/#function evaluations). HNPF's Pareto point density is higher. **NR** - *Not Reported*, **NS** - *Not Supported*, **F** - *Fails*. Remaining Cases in **Appendix I**. The Pareto point density are reported *w.r.t.* an error threshold of $\epsilon = 5e - 4$ in the neighborhood of the true front.

| Case | HNPF | mCHIM | PK | MOOMTL | PMTL | EPO | PHN |
|------|------|-------|-----|--------|------|-----|-----|
| II | 1.83 | 4.39e-2 | NR | 1.00 | 0.05 | 0.05 | 0.05 |
| III | 1.38 | 1.00e-2 | NR | NS | NS | NS | NS |
| IV | 1.34 | 1.38e-6 | 2.22e-4 | F | F | F | F |
| V | 6.57 | 5.86e-3 | NR | NS | NS | NS | NS |
| VI | 0.20 | NR | 1.32e-2 | NS | NS | NS | NS |
| VII | 6.57 | NR | 9.29e-5 | F | F | F | F |

**MTL Approaches:** One of the main concerns regarding MTL solvers is that a termination criterion associated with the optimization approach is not described. For any method to be called a solver a termination criterion is necessary since the number of iterations to reach a solution point is not known a-*priori*. We suggest the reader avail themselves of the open-source codes provided by the respective MTL authors wherein the solver uses a `for` loop with a pre-specified number of iterations. Furthermore, it is not clear as to how MTL solvers, relying upon a linear scalarized form (SOO) of the original MOO problem, arrive at the non-stationary points of this SOO. Even if the domain is partitioned into sub-domains (cones, rays *etc.*) the stationary points of the SOO remain unchanged. The reader is referred to **Appendix B** for a detailed discussion using a benchmark problem.

**HNPF:** Since HNPF Stage-1 approximates the true Pareto manifold $M(X^*)$, it has the advantage of learning $\tilde{M}(\tilde{X})$ from only 10k training points. No other enhanced scalarization method (mCHIM, PK or MTL) can approximate the Pareto manifold itself, rather detect Pareto points only: hypothetically having just the Fritz-John discriminator in Stage-1 without the neural network preceding it. Additionally, HNPF explicitly uses the Stage-2 filter necessary to remove dominated point from the Stage-1 weak Pareto solution set that cannot be addressed by an optimization problem.

**Results on Multi-MNIST:** If HNPF or OR machineries are correct, then why not show results on practical datasets? This is a valid question since the core objective of any approach is to resolve a practical problem. We are currently faced with the following obstacles: **i)** HNPF and OR methods are not yet neural scalable, hence cannot handle variable dimensions at orders $\sim 10^3$ or higher; **ii)** MTL methods do not have a termination criteria and during numerical evaluation either produce non-Pareto points or fail completely for some of the benchmark non-convex cases. This raises serious concerns regarding their veracity on practical datasets where the ground truth Pareto solution is unknown; and **iii)** Framing Multi-MNIST as a Pareto optimality problem raises serious concerns from first principles, as to how the altered network architecture still abides by the Pareto framing. Readers are referred to **Appendix E** for a detailed discussion on this argument.

## 5.7 NUMERICAL CORRECTNESS OF HNPF

We numerically verify the proposed error bounds for the benchmark cases cosnidered. Since the analytical form of the Pareto solution manifold $M(X^*)$ is

Table 3: Mean Squared Error of HNPF extracted manifold $\tilde{M}$ *w.r.t.* the true analytical manifold $M$.

| Cases | I | II | III | IV | V | VI | VII |
|---|---|---|---|---|---|---|---|
| Error ( $\times 10^{-4}$ ) | 1.2 | 1.2 | 3.7 | 4.1 | 2.5 | 3.2 | 1.7 |

known, we can numerically verify the network approximated manifold $\tilde{M}(\tilde{X})$. **Table 3** shows the $L_2$ error between true and approximated manifolds. Note that for all the experiments since $\epsilon = 5 \times 10^{-4}$, the errors are bounded above by $\epsilon$, even when the functions and constraints are non-convex.

Since we are considering non-convex problems in an effort to edge closer to practical applications, providing theoretical guarantees is not possible without making debilitating convexity assumptions on the functions and constraints. MTL methods are supported by several theorems that hold true only for convex functions wherein numerical results are supplied for non-convex problems. Further, no numerical experiments are presented in MTL works that confirm their theoretical results. On the other hand, HNPF is a practical Pareto optimal set extractor with numerically verifiable error bounds without introducing impractical assumptions.

## 6 CONCLUSION AND FUTURE WORK

A two-stage, neural-filter (HNPF) based optimization framework is presented for extracting the Pareto optimal solution set for MOO constrained optimization problems. HNPF is computationally efficient and scales well with increasing dimensionality of design variable space, objective functions, and constraints *w.r.t.* OR methods. Results on verifiable benchmark problems show that our Pareto solution set accuracy compares well against known solutions for a wide variety of benchmark problems. The benchmark problems are chosen to test out different aspects of our HNPF framework that might arise in practical MOO problems. The proposed neural architecture is interpretable with an FJC guided discriminator for *weak* Pareto manifold classification with an efficient Pareto filter to extract an optimal Pareto set. We also show that the approximation error between the true and extracted Pareto manifold can be easily verified for analytical solutions. In our future work, we will develop an FJC guided solver for a scalable HNPF algorithm to address high-dimensional neural MOO problems.

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
