# OpenReview forum: "A Two-Stage Neural-Filter Pareto Front Extractor and the need for Benchmarking"
_ICLR.cc/2022/Conference — ICLR 2022 Submitted_

### Official Review · Reviewer_yaU2 · 2021-10-29

**Correctness:** 2
**Technical Novelty And Significance:** 2
**Empirical Novelty And Significance:** 2
**Recommendation:** 3
**Confidence:** 4

**Main Review:**

My main concern about this work is its scalability. At first phase of finding the weak pareto front HNPF needs to sample the solution space in uniform manner. Covering the solution space for real life solution is not feasible (in terms of memory and runtime) as such models contain hundreds of millions of parameters. Next, the authors propose a Neural Net (NN) to approximate the weak pareto front, since the NN optimization process is depend on labels there is additional labeling process. Although, this labeling process is automatically (i.e. without human intervention) it requires the calculation of $det(L^TL)$ where $L\in\mathbb{R}^{\(n+m\)\times\(k+m\)}$ per sampled solution. The NN network is working on the solution/model parameter space which is not scalable as NNs contain much more parameters than their input space to ensure degrees of freedom.

For the second step HNPF utilizes variant of Kd trees to filter our dominated solutions. In my perspective the novelty of this section is limited, it can be applied to any MTL/MOO approach. It will be interesting to see how the compared methods' results change after applying this filter.

The experiment part is a bit weak. Due to the scalability problem the authors did not test HNPF on more challenging datasets (not even MNIST).




**Summary Of The Paper:**

The authors present key shortcomings of MTL solvers in addition to Hybrid Neural Pareto Front (HNPF) that aims to handle non-convex functions and constraints.

The authors claim the following contributions:
- New strategy for weak Pareto front identification based on Fritz-John conditions.
- New Pareto filter to remove dominated points from the weak Pareto front.


**Summary Of The Review:**

Some questions were raised during the review process.

---

> ### Author Response · Authors · 2021-11-19
> **Comments for Reviewer yaU2**
>
> Please see the overall comments section that clarifies what possible design choices are needed for a Pareto solver.
>
> 1. My main concern about this work is its scalability. ...contain hundreds of millions of parameters.
>
> We agree with the concern of the reviewer that the current HNPF work is not scalable to neural setting. We ourselves have highlighted it in Table 1. However, note that accuracy comes before scaling for any solvers. Current MTL solvers are failing on the basic non-convex benchmark cases, however are scaled for practical problems. Where's the guarantee that they are even producing something remotely close to the Pareto front for those neural problems? Note that if all of MOOMTL, MTL, EPO and PHN are "correct", how are they producing different fronts for the same Multi-MNIST dataset using the same LeNet?
>
> 2. Next, the authors propose a Neural Net (NN) ... than their input space to ensure degrees of freedom.
>
> Please note that once the functions and constraints are defined, Tensorflow uses autograd to save a a calculation of $det(L^TL)$ matrix, which is then evaluated at runtime for different values of $x$. We would like to point out that MTL approaches follow the same procedure too, they are just restricted to functions and cannot handle constraints.
>
> 3. For the second step HNPF utilizes variant of Kd treesresults change after applying this filter.
>
> Yes, the Filter can be used by any MTL method too. However, shockingly, we found that none of the MTL methods even talk about such a filter. Note that unless all the weak Pareto points are gathered, one cannot apply the filter to remove certain points. It is akin to saying that unless one collects all the minimas over a non-convex loss surface, one cannot truly identify the global minima, because the gradient is zero at all minimas.
>
> 4. The experiment part is a bit weak. Due to the scalability problem ... more challenging datasets (not even MNIST).}
>
> We again point out that we only scale an algorithm if it is correct in the first place. Currently so called "MTL Pareto solvers" fail for most of the benchmark scenario we have tested on. However, they are being scaled to large problems. Note that the ground truth Pareto front for MNIST is unknown. Hence, we would like a to ask which MTL Pareto solver is correct, since for the same MNIST dataset with Le-Net different MTL methods have produced different Pareto front.
>
> Finally we would like to point the reviewer to the code attached in the Supplemental material where we have given simple codes and geometric arguments of why and how MTL methods are failing for cases considered in this work.

---

### Official Review · Reviewer_gpGn · 2021-11-01

**Correctness:** 2
**Technical Novelty And Significance:** 2
**Empirical Novelty And Significance:** 1
**Recommendation:** 3
**Confidence:** 5

**Main Review:**

Although they are claiming that their proposed algorithm solves the issues mentioned in the paper, I think the literature is not represented fairly to show novelty here. For instance, they mentioned that "the theorems presented in MTL works rely upon convexity assumption that are never justified through numerical experiments." This claim is not true. Just for one case of [A], they explicitly discuss that they have no convexity assumptions. There are other proposals, mostly with multiple gradients descent algorithm (MGDA) variations that have convergence guarantees for both convex and non-convex objectives such as [B]. Besides, there is confusion regarding the convexity, whether they are discussing the convexity of objective functions or the Pareto frontier. Since even with convex objective functions, the Pareto frontier can be still non-convex.

In addition to the lack of fair representation of the literature to justify the claims, I have several other concerns regarding the approaches in this paper as follows:
1. In most parts, they are using algorithms and proposals from other papers, hence it is not clear what is their own novelty added in this paper. For instance, the FJC for the optimality and Kd-tree algorithm seems to be borrowed from other papers. Hence, it is not clear, what is the novelty in this paper since these are the main parts of this proposal.  They use the optimality condition derived from FJC to use as a loss function for an MLP in order to decide whether a point is a Pareto or not.

2. In addition, as they mentioned themselves, their proposal has some similarities with PHN [C]. In the case of PHN, the algorithm and training procedure is an end-to-end solution and very clear. However, in this paper, the training procedure is not clear at all. The main concern is that how they are getting the $X$ parameters in the first place to feed to their MLP model for prediction of Pareto and not Pareto. Are they using some existing methods like EPO to find $X$? The training procedure should be stated clearly.

3. The filtering of Pareto points is not novel in my opinion. This is the procedure most of the proposed algorithms in this domain are doing one way or another. Since this is a search space over candidates for dominated points on the weak Pareto set is not that big, the computational complexity is not that big a concern to be claimed as the novelty here. See this [link](https://stackoverflow.com/a/40239615) for several solutions for this problem in the community.

In addition to the aforementioned concerns, one of the main concerns, I reflected upon several times, is regarding the clarity of the manuscript. I believe many parts of this paper require more clarification and in-depth discussion while some parts can be omitted or moved to the appendix. The distinction between the contribution of this paper and other proposals should be clear, and the contribution of the literature should addressed fairly and clearly.



[A] Debabrata Mahapatra and Vaibhav Rajan. Multi-task learning with user preferences: Gradient descent with controlled ascent in pareto optimization. In International Conference on Machine Learning (ICML), pp. 6597–6607, 2020.

[B] Kamani, Mohammad Mahdi, et al. "Pareto Efficient Fairness in Supervised Learning: From Extraction to Tracing." arXiv preprint arXiv:2104.01634 (2021).

[C] Aviv Navon, Aviv Shamsian, Gal Chechik, and Ethan Fetaya. Learning the pareto front with hypernetworks. In International Conference on Learning Representations (ICLR), 2021. URL https://openreview.net/forum?id=NjF772F4ZZR.

**Summary Of The Paper:**

This paper intends to find points on the Pareto frontier of a multiobjective optimization problem. Despite other baseline methods for this problem, they claim that their proposed algorithm is suitable for non-convex optimization, and their solutions are evenly spread across the frontier. Also, their algorithm can handle constraints for the MOO problems similar to solutions proposed in existing operation research methods. Empirical evaluations show that the proposed algorithm can find Pareto points and omit non-dominated points in the final stage.

**Summary Of The Review:**

Overall, I believe the novelty of the proposed algorithms is marginal and the clarity of the paper is very low at this stage.

---

> ### Author Response · Authors · 2021-11-19
> **Comments for Reviewer  gpGn**
>
> Please see the overall comments section that clarifies what possible design choices are needed for a Pareto solver.
>
> 1. Although they are claiming that ... (MGDA) variations have guarantees for both convex and non-convex objectives.
>
> The main crux of our work is not around novelty but the correctness of the solution. The benchmark cases are not designed by us but the MTL solvers fail/crash completely which hampers our confidence. Our primary objective was not to beat the existing algorithms but to use them for our specific application wherein these claimed solvers unfortunately did not succeed. A claim of convergence guarantees does not imply that these approaches will work in practice.
>
> For EPO (as well as for other MTL methods who made that claim), we really wanted them to work since it would have made our life easier. However, the numerical experiments using their provided codes lead us to a different conclusion, something that is not evident from the theorems provided.
>
> The reviewer is referred to page 13 of [B], Assumption 2a, Lemma 2, Lemma 3 and Theorem 2, where strong convexity assumptions are made to guarantee convergence with some error tolerance. Without making assumptions on the non-convexity of the problem itself, no theorems or lemmas can be written. Furthermore, the theorem on non-convex functions [B] depends on all of the aforementioned convexity and smoothness assumptions, which none of the considered datasets in their work satisfy.
>
> 2. Besides, there is confusion regarding the convexity, ... the Pareto frontier can be still non-convex
>
> A plot of f1 vs f2 can visually be convex or non-convex and does not imply that f1 is a convex (or non-convex) function of f2 (or vice versa) since they are separate functions of x. As such the visual convexity (or non-convexity) of the Pareto front on an f1 vs f2 plot is of no consequence. To solidify this remark and resolve any confusion, let us consider the following two cases:
>
> Case 1: f1(x) = (x-1)^2, f2(x) = (x+1)^2
> The two functions are convex in x and visually the Pareto front for a plot between f1 and f2 is convex.
>
> Case 2: See Case IV in the numerical experiments section of the main text. Although, the two functions f1 and f2 are non-convex (sinusoidal) in x, the Pareto front is still visually convex.
>
> 3. In most parts, they are using algorithms and proposals from other papers.. to decide whether a point is a Pareto or not.
>
> The FJC criteria is certainly borrowed from OR literature (Gobbi 2015). However, if the reviewer notices carefully, Gobbi just used the FJC to handle convex objectives only. We are extending the FJC concept to serve as a neural discriminator for Stage 1 HNPF, that is now supporting both non-convex functions and constraints. We are thus confused as to how this is a mere copy of the previous work.
>
> Kd-trees are used for efficient space partitioning. We never used Kd-Trees in our algorithm, rather our filter is inspired by them. We are confused as to how this is a mere copy. The work of EPO borrows the entire concept of tracing rays wrt utopia point from NBI. So we are confused as to the novelty of EPO, when it is failing on simple benchmark cases where NBI succeeded, but somehow scale to high dimensional problems and "hopefully" gives the correct answer.
>
> 4. In addition, their proposal has some similarities with PHN [C]. ...training procedure should be stated clearly.
>
> We have a section allocated to similarities and differences between HNPF and PHN, since PHN is our competitor method (ICLR 2020). Please refer to Section 4.3 and Appendix D to see how PHN fails when the front is discontinuous or sparse, or when EPO fails to converge.
>
> We would also like to raise a question to the reviewer: what is your opinion as to whether LS works or not? We ask this question because all MTL works (including PHN) have claimed that LS does not work for solving Pareto, hence the need for their specialized MTL solvers. However, please note that PHN uses both EPO and LS to solve for high-dimensional neural problems. So if LS does not work, then how does PHN-LS work?
>
> 5. The filtering of Pareto points is not novel in my opinion.
>
> Is "Stack Overflow" link is the standard of the ICLR? The Stage 2 filter is developed solely for the fact that the original Pareto filter is of complexity O(P!), for P Pareto points. Certainly if P is low (as in the case of MTL algorithms producing five points for a problem), the base filter can be applied. However, a theoretical complexity of an algorithm is pursued, specially when the number of weak Pareto points are large.
>
> 6. In addition .. contribution of the literature should addressed fairly and clearly.
>
> Refer to Table 1 in main script for contribution and comparison against existing literature. If the reviewer feels that parts needs reorganizing, we are happy to do so. Refer to the overall comments to get a sense of the message that we wanted to communicate to the community, which might not be clear in writing.

---

### Official Review · Reviewer_pPPS · 2021-11-02

**Correctness:** 2
**Technical Novelty And Significance:** 2
**Empirical Novelty And Significance:** 1
**Recommendation:** 3
**Confidence:** 4

**Main Review:**

**Strengths:**

+ It studies how to find the whole Pareto set for multi-objective optimization problems, which could be useful for different applications.

+ The call for new benchmarks for multi-objective optimization other than multi-task learning is valuable, also see a related work [1] (I am not involved in this paper). However, this discussion is in the appendix and not the main point of this work.

**Weakness:**

*Method:*

**1. Verification and Optimization:**

The proposed HNPF method is for verification (e.g., check whether a given solution x is Pareto optimal), but not for optimization (e.g., find an (approximate) Pareto solution x). It needs an extra search method, such as random sampling in this work, to first generate a large number of feasible solutions to cover the whole search space. Therefore, the underlying optimization is indeed random sampling (independent from HNPF), which could be extremely inefficient for a non-trivial search space. It is not suitable to put and compare the proposed HNPF method with other optimization methods that can directly find the (approximate) Pareto solution.

Since HNPF depends on random sampling, it is not surprising that it can only work for small scale problems.

**2. The Reason to Build the Model:**

HNPF needs to first build a neural network to check whether a given solution x satisfies the Fritz-John Condition (FJC), which requires a large number of training samples (e.g., 11K for a two-dimensional problem). The learned model is mainly used to classify whether the extra randomly sampled solutions (e.g., 9K) are weak Pareto optimal or not. The reason for model building, such as the advantage over the simple FJC rule-based classification, is not well motivated and justified in this work.

The proposed Pareto filter in stage 2 is not discussed and compared with other related nondominated sorting algorithms (e.g., [2]).

**3. Necessary Condition for Pareto Optimality:**

The KKT[3] and FJC[4] are two types of first order necessary conditions for (local) Pareto efficiency (Pareto optimality). In my understanding, the multi-objective optimization based MTL algorithms mentioned in this work (Sener & Koltun, 2018; Lin et al., 2019a; Mahapatra & Rajan, 2020; Ma et al., 2020; Navon et al., 2021) mainly use the gradient-based multi-objective optimization methods (e.g., MGDA) [5-7], which is based on the KKT condition. For these methods, in each update step, the gradient can be written as a linear combination of the gradient for each objective with adaptive weights derived from the KKT condition. Therefore, they are all different from the simple linear scalarization with fixed weights. All the claims and analyses in this work that the previous works use simple linear scalarization is not correct.

The FJ condition is also for local Pareto convergence, similar to the KKT condition. The global convergence property is solely due to random sampling that only works for extremely low-dimensional problems. It is unfair to say the proposed algorithm can overcome the local convergence of other gradient-based methods. In addition, the proposed algorithm heavily depends on the Fritz-John condition, but the original work [4] is not cited.

**4. Linear Scalarization and Convex Pareto Front:**

It is well-known that the simple linear scalarization cannot find the non-convex part of the Pareto front [8]. This finding leads to the seminal work on NBI scalarization (Das & Dennis.,1998), which is indeed one fundamental work that inspires the proposed method in this work (section 4, first sentence). The claim "it is incorrect to state that LS itself fails if the Pareto front is non-convex" (appendix, page 15) is questionable.

Since the proposed HNPF can only verify whether a given solution is weak Pareto optimal, its ability to find the whole Pareto front totally depends on the extra sampling method (e.g., random sampling) to generate all the Pareto solutions (might be infinite). It is misleading to indicate the proposed HNPF method itself can find the whole Pareto front. In addition, since the Pareto set has measure zero and infinite cardinality, the random sampling + HNPF method can at most find a dense approximation to the Pareto set.

*Experiment:*

**5. Algorithms for Comparison:**

All the multi-objective optimization based MTL algorithms are designed for optimizing a deep neural network with millions of parameters. They implicitly depend on the assumption that the deep neural network has good properties (e.g., no bad local optimum [9][10]) on its loss functions, which is consistent with other gradient-based single-objective optimization methods. They are not designed to find the global Pareto front for low-dimensional problems.

For low-dimensional problems, it is more suitable to compare with the model-free multi-objective optimization methods such as the multi-objective evolutionary algorithm [11,12] and multi-objective CMA-ES [13]. If the model building is allowed, Multi-Objective Bayesian Optimization (MOBO) algorithms can have a very good sampling efficiency for the low-dimensional problems [14,15]. It is also very common to conduct non-dominated filtering at the end of those model-free algorithms or MOBOs (e.g., only keeping the current non-dominated solutions).

**6. Training + Sampling:**

The proposed method needs to first sample 11k solutions to train the neural network model, then randomly generate extra 9K solutions for filtering. Is there any advantage over simply using FJC to filter 9K (or 11k + 9K) randomly sampling solutions?

**7. Figure from Other Works:**

Many figures in the main paper and the appendix are directly borrowed from other works. I think this is not appropriate even the credits are given to the original works.

**Reference:**

[1] Ruchte, Michael, and Josif Grabocka. Multi-task problems are not multi-objective. arXiv preprint arXiv:2110.07301, 2021.

[2] Roy, Proteek Chandan, Kalyanmoy Deb, and Md Monirul Islam. An efficient nondominated sorting algorithm for large number of fronts. IEEE transactions on cybernetics 49, no. 3: 859-869, 2018.

[3] Kuhn, H. W., and A. W. Tucker. Nonlinear Programming. In Proceedings of the Second Berkeley Symposium on Mathematical Statistics and Probability, pp. 481-492. University of California Press, 1951.

[4] Da Cunha, N. O., and E. Polak. Constrained minimization under vectorvalued criteria in finite dimensional spaces. Journal of Mathematical Analysis and Applications, 19(1), 103–124 ,1967.

[5] Fliege, Jorg, and Benar Fux Svaiter. Steepest descent methods for multicriteria optimization. Mathematical methods of operations research 51, no. 3: 479-494, 2000.

[6] Fliege, Jorg, and A. Ismael F. Vaz. A method for constrained multiobjective optimization based on SQP techniques. SIAM Journal on Optimization 26, no. 4: 2091-2119, 2016.

[7] Desideri, Jean-Antoine. Multiple-gradient descent algorithm (MGDA) for multiobjective optimization. Comptes Rendus Mathematique 350, no. 5-6: 313-318, 2012.

[8] Das, Indraneel, and John E. Dennis. A closer look at drawbacks of minimizing weighted sums of objectives for Pareto set generation in multicriteria optimization problems. Structural optimization 14, no. 1: 63-69, 1997.

[9] Kawaguchi, Kenji. Deep learning without poor local minima. NeurIPS 2016.

[10] Kawaguchi, Kenji, and Leslie Kaelbling. Elimination of all bad local minima in deep learning. AISTATS 2020.

[11] Deb, Kalyanmoy, Amrit Pratap, Sameer Agarwal, and T. A. M. T. Meyarivan. A fast and elitist multiobjective genetic algorithm: NSGA-II. IEEE transactions on evolutionary computation 6, no. 2: 182-197, 2002.

[12] Zhang, Qingfu, and Hui Li. "MOEA/D: A multiobjective evolutionary algorithm based on decomposition." IEEE Transactions on evolutionary computation 11, no. 6: 712-731, 2007.

[13] Igel, Christian, Nikolaus Hansen, and Stefan Roth. Covariance matrix adaptation for multi-objective optimization. Evolutionary computation 15, no. 1: 1-28, 2007.

[14] Daulton, Samuel, Maximilian Balandat, and Eytan Bakshy. Differentiable Expected Hypervolume Improvement for Parallel Multi-Objective Bayesian Optimization. NeurIPS 2020.

[15] Konakovic Lukovic, Mina, Yunsheng Tian, and Wojciech Matusik. Diversity-Guided Multi-Objective Bayesian Optimization With Batch Evaluations. NeurIPS 2020.



**Summary Of The Paper:**

This work proposes a Hybrid Neural Pareto Front (HNPF) framework to solve multi-objective optimization problems. The proposed method needs to first generate a huge number of feasible solutions to cover the whole decision space (e.g., random sampling in this work), and then use a two-stage approach to select the Pareto optimal solutions. In the first stage, it builds a Fritz-John Condition (FJC) based neural network to verify the weak Pareto optimality for given solutions. In the second stage, it further filters the truly strong Pareto optimal subset from the obtained solutions. Experiments have been conducted on several low-dimensional problems to validate the proposed method's performance.


**Summary Of The Review:**

This work aims at finding the Pareto front for a multi-objective optimization problem, which is important for many applications. However, there are many major concerns on both the method (verification rather than optimization, the reason for model building, questionable claims) and experiments (suitable comparison, figures from other works) that make this work unacceptable in the current form.

============================= post rebuttal response =============================

I have read the other reviewers' comments with the authors' feedback, but still think the proposed verification method (FJC discriminator) should not be directly compared with other optimization methods (not limited to MTL). Therefore, I keep my initial score (3) and lean toward rejection.

---

> ### Author Response · Authors · 2021-11-19
> **Comments for Reviewer pPPS**
>
> Please see the overall comments section that clarifies what possible design choices are needed for a Pareto solver.
>
> 1. The call for new benchmarks
>
> We thank the reviewer for pointing us to this new arxiv paper. We are stating the same idea in Appendix E (Issue with Pareto framing of MTL networks). While the arxiv paper demonstrated it empirically through numerical experiments, we present the same contradiction from algebraic and geometric principles. The section can be rearranged if needed.
>
> 2. Verification and Optimization
>
> We agree with the reviewer completely that the current method is just a discriminator and as such is not efficiently scalable to higher-dimensional neural problem. However, our intent with this work was not scalability but to design an FJC guide discriminator that can be used to design an efficient solver later. Most of the recent MTL solvers (although claimed to be efficient, correct and scalable) are either not converging or failing completely for some of the simplest cases (Table 2 main text). We therefore focused our effort on first obtaining convergence (termination criterion) before ramping to high dimensional problems.
>
> 3. The Reason to Build the Model:
>
> The FJC discriminator is a classifier that separates Pareto optimal from non-optimal points using an initial random set of 11K points. The neural network converges to the Pareto front with a user-prescribed error tolerance. After training the network efficiently stores the Pareto front as an indicator function (1 for Pareto and 0 for non-Pareto optimal) on the variable domain. The non-dominated filter (Stage 2) then runs on a reduced set (obtained from Stage 1) to remove dominated points. The main thrust is that the none of the MTL works consider the fact that an optimization based solver (EPO or otherwise) can still contain dominated Pareto points as shown in our numerical experiments and therefore a filter is necessary.
>
> 4. Necessary Condition for Pareto Optimality:
>
> We humbly disagree with the reviewer that the aforementioned methods do not rely upon LS. Please see the overall comments with a discussion on how multiple-objective functions are used to obtain a LS objective. Note that an optimization cannot be performed over a vector objective since the termination (convergence) criterion cannot be established. Thanks for the citation on FJC.
>
> 5. LS and Convex Pareto Front:
>
> LS is the process of obtaining a single scalar objective function by linearly combining multiple objectives. The process itself cannot be blamed to fail since the choice of minimization or maximization over the LS function is user dependent. As shown in Appendix B, LS works just fine if slices along alpha are chosen followed by maximization of the scalarized function restricted to a specific choice of alpha.If we minimize the LS function instead of maximization we will only obtain the two end-point minima as is evident from Fig. 9 for different slices in alpha. NBI avoids this issue of choosing  between minimizing by simply finding points where the gradient becomes close to zero along a ray originating from an ideal/utopia point.
>
> 6. Algorithms for Comparison:
>
> Note that Kawaguchi (NIPS 2016) states that: "We note that even though we have advanced the theoretical foundations of deep learning and non-convex optimization, there is still a gap between theory and practice." The authors have not found any neural classifier or regressor that has been shown to have a convex loss landscape by design especially when we consider the multi-MNIST problem or for Le-Net. None of the previous authors have shown that the classifier loss function is convex and therefore assuming convexity is not valid.
>
> It would be incorrect to state that MTL works are not designed to solve low-dimensional benchmark problems while at the same time are accurate and efficient for high dimensional problems. For e.g., an iterative linear solver works equally well for a 2x2 matrix or a 1Mx1M matrix both in terms of accuracy and efficiency, given a solution exists. A target that all robust and efficient solvers must satisfy. Note that all MTL works provide a scalable benchmark using two inverted Gaussian objectives in both low and high-dimensional variable space to show scalability and correctness. Our primary concern is that more benchmark cases must be considered other than Gaussian functions.
>
> 7. The proposed method needs to first sample .. randomly sampling solutions?
>
> The Pareto front is efficiently stored as an indicator function by the neural regressor and the additional points are only generated to visually demonstrate to the reader that the front has been accurately identified and stored. Note that the network is trained on 11k points and tested on 90k (not 9k) to ensure that the learned Pareto front is not exhibiting any oscillatory artifacts due to overfitting. The user can choose to plot the surface with any density they want.

---

> > ### Comment · Reviewer_pPPS · 2021-11-29
> > **Thank you for the detailed response**
> >
> > Thank you for the detailed response. I have read the other reviewers' comments with the authors' feedback, but still think the proposed verification method (FJC discriminator) should not be directly compared with other optimization methods (not limited to MTL). Therefore, I keep my initial score (3) and lean toward rejection.

---

> > > ### Author Response · Authors · 2021-11-29
> > > **Response to pPPS**
> > >
> > > We are thankful for the reviewer's response. Since all the methods in this area are targeting the same solution (Pareto Optimality) our intent with the comparison is to indicate a need for setting up benchmarks that can be verified by the user.
> > >
> > > It came to us as a surprise that solvers that are shown to work for high dimensional neural problems are outright failing for some of the simplest test cases.
> > >
> > > We understand the reviewers' decision but if we may ask: what is the most glaring area that can be improved next time? Based upon the reviews we agree that scaling to higher dimensional problems with HNPF is limited.

---

### Official Review · Reviewer_GeVa · 2021-11-02

**Correctness:** 3
**Technical Novelty And Significance:** 2
**Empirical Novelty And Significance:** 3
**Recommendation:** 6
**Confidence:** 3

**Main Review:**

Pros:
- I resonate with the call for analytical benchmarks. Multi-objective optimization as a field may be growing within ML, but I feel many work are not evaluated in a way that is conducive to a holistic understanding of the methods capabilities and limitations. Using these benchmarks, the paper establishes clear cases where recent method fails.
- The paper helps bridge some of the gap between multi-objective work in ML and Operations Research.  I think an ML audience will learn much from the paper, just like I did.
- The proposed two-stage method is reasonable, and shows good performance on the proposed benchmarks.

Cons:
- The proposed two-stage method is a somewhat orthogonal contribution, and I am not sure if it really connects well with the benchmark contribution.
- I wonder if the neural net for weak Pareto front will be practical in applications where it is difficult to compute the value in objective space. For example, if we were to optimize the hyperparameters of deep neural nets like [1] and [2], it may be expensive to obtain samples for this neural net. Some discussion would be good.
- Pareto density (Table 2) is not the only way to evaluate, and it does penalize methods like MTL. There are other evaluation metrics, like hyper volume, which may result in a different ranking of methods. Some discussion would be desired.

[1] Zhang & Duh. Reproducible and Efficient Benchmarks for Hyperparameter Optimization of NMT Systems
. TACL2020 https://aclanthology.org/2020.tacl-1.26/
[2] Klein & Hutter. Tabular benchmarks for joint architecture and hyper-parameter optimization. https://arxiv.org/abs/1905.04970

Additional comments:
- It will be great if the authors release the data or generator code (and evaluation code) for the benchmarks in an easy-to-use fashion. I understand it is relatively easy to implement, but it can help encourage more people to work on the benchmark.

**Summary Of The Paper:**

This paper has two contributions. First it argues for the need to benchmark multi-objective problems  using analytic functions for convex and non-convex conditions. Second, it presents a two-stage method for finding the Pareto front.

**Summary Of The Review:**

I think this paper presents a strong argument for benchmarks in multi-objective problems. More discussion that ties the benchmark and the evaluation of methods back to various practical considerations (e.g. feasibility of sampling in the objective space, desirability of Pareto density vs. hypervolume) would make this paper even stronger.

---

> ### Author Response · Authors · 2021-11-19
> **Comments for Reviewer GeVa**
>
> Please see the overall comments section that clarifies what possible design choices are needed for a Pareto solver.
>
> 1. "I resonate with the call for analytical benchmarks. ... not sure if it really connects well with the benchmark contribution.}
>
> We appreciate the reviewers' comments. Our initial intent was to simply use one of the existing approaches to solve a problem relevant to us. However, while testing these solvers (publicly available Github codes) we found several issues that eventually led us to bench-marking and design of a FJC guided discriminator for evaluation. These issues include:
> - Absence of termination criterion against a prescribed tolerance precluding numerical convergence tests.
> - A solver usually relies on a termination criterion (expected error tolerance) since it is not known how many iterations will the problem take to converge. MTL solvers use for-loops with pre-prescribed number of iterations (around 200) expecting this number to hold for every problem. Please refer to EPO's github code (https://github.com/dbmptr/EPOSearch/blob/master/toy\_experiments/solvers/) for the solver codes in Line 20 (moo\_mtl.py) and Line 91 (pmtl.py), regarding \textit{for loop} usage for gradient descent without any termination criteria whatsoever.
> - EPO has a tolerance criteria in Line 36 (epo\_search.py), however, $\alpha$ in the code is constant, and does not change as delineated in the original EPO \citep{mahapatra2020multi} algorithm (Line 7 Algorithm 1: Find $\beta$ by solving LP (Eq 24)). One can refer to Line 24 (alpha = lp.get\_alpha(l, G, relax=relax)) in epo\_search.py where the function calculates a value of $\alpha$ (or $\beta$ in the algorithm). However, in the actual function get\_alpha(), one can see that Line 66 (in epo\_lp.py) returns a value of alpha, where the function never ever does any modification on the value of alpha based on the input arguments l, G, passed onto the function itself.
> - MTL solvers crashing completely for some of the simplest, low dimensional, non-convex objective functions while claiming applicability to high-dimensional neural networks.
> - Inability to handle user-prescribed non-linear constraints and restrictions on feasible domain.
>
> Currently, we do not claim to be scalable (Table 1 main text) but are able to satisfy a user prescribed error tolerance on verifiable benchmark problems.
>
> 2. "I wonder if the neural net for weak Pareto front will be practical in applications where it is difficult to compute the value in objective space. For example, if we were to optimize the hyperparameters of deep neural nets like [1] and [2], it may be expensive to obtain samples for this neural net. Some discussion would be good."
>
> We have already developed an extension of our current HNPF framework which scales well for high-dimensional problems while satisfying all the benchmark problems within a user prescribed tolerance. However, we considered getting feedback from the community as to what can constitute as a robust solver moving forward. We cannot mention the name of the work yet due to double blind concerns.
>
> 3. "Pareto density (Table 2) is not the only way to evaluate, and it does penalize methods like MTL. There are other evaluation metrics, like hyper volume, which may result in a different ranking of methods. Some discussion would be desired.}
>
> We agree with the reviewer. However, note that a user-prescribed error tolerance is necessary to control a solver for comparison on a common ground. Since the ground truth solution is known for these benchmark problems, the errors can be evaluated directly either locally or globally. Through numerical experiments we show (in main text and appendix) that most of the MTL methods either converge partially (see Fig. 4 where the errors are visually discernible) or fail completely for our benchmark problems (Table 2 main text). An issue with the HyperVolume (HV) is that it is evaluated with respect to a utopia/ideal point wherein the HV measure can become small when the computed Pareto front is closer to the utopia/ideal point while being substantially off from the true front.

---

> > ### Comment · Reviewer_GeVa · 2021-11-29
> > **Just asking for discussion**
> >
> > Thanks for the response. Regarding cons points 2 & 3, I want to clarify that I'm just asking for additional discussion in the paper. For example, I'm not asking the paper to include a more scalable method, but simply asking for a paragraph in the paper that discusses potential limitations due to scalability. Similarly, for the point about Pareto density, I'm just asking for a discussion that acknowledges other evaluation methods. This is mainly to make sure the audience gets a broader picture and can properly contextualize the results in the paper.

---

> > > ### Author Response · Authors · 2021-11-30
> > > **Response for GeVa**
> > >
> > > We appreciate the reviewers response and as mentioned in the main text (Table 1) shown the current scalability which can be prohibitive for large neural problems.
> > >
> > > However, we want to ask the question: is scaling a method more important when the method is not able to generate accurate results   or outright crash for simple cases.
> > >
> > > We went through the codes of some of all of the compared solvers and found serious issues as mentioned  in the previous response.

---

### Author Response · Authors · 2021-11-19
**Overall Comments on Pareto Optimality**

# Pareto Optimality and Stationary Points

1. Consider two multi-variable functions (for ease of description). Let us consider the following two quadratic functions (convex in both variables): $f_{1}(\mathbf{x}) = (x_{1}-1)^{2} + (x_{2}-1)^2$ and $f_{2}(\mathbf{x}) = (x_{1}+1)^2 + (x_{2}+1)^2$
2. The task is to find a Pareto front between the two objectives. Since this is an easy problem the Pareto front is known analytically as the straight line $x_{1} = x_{2}$ for $x_{1} \in [-1,1]$ in the variable domain.
3. Note that independent of each other $\nabla f_{1}(\mathbf{x}) = \left[\frac{\partial f_{1}}{\partial x_{1}} \quad \frac{\partial f_{1}}{\partial x_{2}}\right]^{T} = \mathbf{0}$ at $(x_{1},x_{2}) = (1,1)$ and  $\nabla f_{2} = \left[\frac{\partial f_{2}}{\partial x_{1}} \quad \frac{\partial f_{2}}{\partial x_{2}}\right]^{T} = \mathbf{0}$ at $(x_{1},x_{2}) = (-1,-1)$ starting from
4. One can also confirm that the gradient matrix $G $ is not identically zero for any value of $x \in \mathbb{R}^{2}$.

# Target: An algorithm starting from a point $(x_{1},x_{2})$ such that ($f_{1},f_{2}$) is in the feasible region to converge (or terminate) at the Pareto front within a user prescribed tolerance.

5. Let us say we have an initialization $\mathbf{x} = [x_{1} \quad x_{2}]$ at iterate $k=0$. The update is:

    $\mathbf{x}^{k+1} = \mathbf{x}^{k} \pm \eta F(\mathbf{x})$ where $\pm$ is used to indicate that a choice of minimization or maximization is not yet decided.
* Requirement 1: Here, $\eta \gt 0$ is the step size (user-prescribed) and the unknown vector function $F(\mathbf{x})$ should be such that $F(\mathbf{x}) = \mathbf{0}$ (termination) at the Pareto candidate points.
* Requirement 2: The vector function $F(\mathbf{x})$ should be somehow related to the original objectives $f_{1}$ and $f_{2}$.

### Note that this iterative update is present (in some form) in all MTL Pareto solvers.

# Choice of $F(\mathbf{x})$ and Convergence
1. An easy choice of $F(\mathbf{x})$ is given by a linear scalarization of the objectives $f_{1}$ and $f_{2}$ to create a single scalar objective function $S(\mathbf{x}) = \alpha_{1} f_{1}(x) + \alpha_{2} f_{2}(x)$
2. The vector update function $F(\mathbf{x})$ is now given by the gradient of $S(\mathbf{x})$ with respect to $x_{1}$ and $x_{2}$, (given $\alpha_{i}$s) as: $F(\mathbf{x}) = G\alpha = [\nabla f_{1}(x) \quad \nabla f_{2}(x)] [\alpha_{1} \quad \alpha_{2}]^{T}$
3. To avoid a trivial solution the vector $[\alpha_{1} \quad \alpha_{2}]$ must also not be identically zero. So $G\alpha$ should approach zero for some $\mathbf{x} = [x_{1} \quad x_{2}]$ as we iteratively update $x$ using $F(\mathbf{x})$.

###   What $\alpha$ values can achieve the above termination criterion?

* We know that $G\alpha=\mathbf{0}$ gives us the stationary points.
* Further any matrix $G \neq \mathbf{0}$, $G\alpha=0$ can be solved for a non-trivial $\alpha \neq 0$ iff G has a null-space; or G is low rank; or if G is square then it's determinant is zero.

### This implies that a stationary point corresponds to $(x_{1},x_{2})$ such that G becomes low rank (Fritz John conditions).

# Uniquely determining $\alpha_{i}$s and adaptive or explicit update of $\alpha_{i}$s

### Let us consider a general gradient matrix G now.

* Case 1: If the gradient matrix $G$ is full rank. Then $\alpha = \mathbf{0}$  (vector is identically zero) is the only solution.
* Case 2: If the gradient matrix $G$ has a rank deficiency $q=1$ (one rank deficient). Then one of the $\alpha_{i}$s can be chosen arbitrarily or only one equation is missing. A simplex criterion $\sum_{i}\alpha_{i} = 1$ then supplies this arbitrary choice.
* Case 3: Here comes the trouble: If matrix $G$ has a rank deficiency $q>1$ (more than one rank deficient) more than one of the $\alpha_{i}$s can be chosen arbitrarily. The simplex criterion is no longer sufficient for a unique $\alpha$ vector.

### $det(G) = 0$ is the most general way to state rank deficiency of $G$ irrespective of the rank of $G$. For a non-square $G$, $det(G^{T}G)=0$ is then equivalent.

# Why does linear scalarization fail for non-convex objectives?

As mentioned in the main text, it is not the process of linear scalarization that fails but the following choice:
* We know that the stationary points correspond to $\mathbf{x}^{*} =$ {$x|det(G(\mathbf{x})=0$}. However, this does not discern a stationary point as minimum, maximum, or saddle point. A gradient optimization requires an explicit description of this choice.
* Since Pareto points are not known a-priori, this choice cannot be made arbitrarily. The reader can confirm that for convex objectives a gradient descent is needed whereas for the numerical experiment in Appendix B, a gradient ascent must be performed.
* In higher dimensions, this choice becomes tedious (ascent/descent direction), $det(G)=0$ now eases the problem.

### Are all the stationary points Pareto optimal? Not necessarily and therefore a Pareto filter is needed.

---

### Decision · Program_Chairs · 2022-01-20

**Decision:**

Reject

**Comment:**

The paper makes two contributions: (1) Multi-task benchmarks where the pareto solution is known analytically; and (2) a verification method for testing if solutions are on the pareto front. The authors make the point that MTL methods are applied to large-scale problems, but fail to find the pareto front in problems where it is known.

Reviewers appreciated the discussion and insights by the approach, and the idea that correctness of scalable methods should be evaluated with problems that have analytic solutions, but they also had grave concerns. The primary concern is that without an efficient search, a verification method that builds on filters randomly generated solutions cannot scale to high dimensional problems. There were also disagreement about the role of LS and comparison with previous literature.

As a result, the contribution of the submission is not sufficient for acceptance to ICLR